# Finding Central Vertices and Community Structure via Extended Density Peaks-Based Clustering

Yuanyuan Meng [1,2] and Xiyu Liu [1,*]

1   Academy of Management Science, School of Business, Shandong Normal University, Jinan 250014, China; mengyy@sdufe.edu.cn
2   School of Computer Science and Technology, Shandong University of Finance and Economics, Jinan 250014, China
*   Correspondence: xyliu@sdnu.edu.cn

**Abstract:** Community detection is a significant research field of social networks, and modularity is a common method to measure the division of communities in social networks. Many classical algorithms obtain community partition by improving the modularity of the whole network. However, there is still a challenge in community division, which is that the traditional modularity optimization is difficult to avoid resolution limits. To a certain extent, the simple pursuit of improving modularity will cause the division to deviate from the real community structure. To overcome these defects, with the help of clustering ideas, we proposed a method to filter community centers by the relative connection coefficient between vertices, and we analyzed the community structure accordingly. We discuss how to define the relative connection coefficient between vertices, how to select the community centers, and how to divide the remaining vertices. Experiments on both real and synthetic networks demonstrated that our algorithm is effective compared with the state-of-the-art methods.

**Keywords:** community detection; node similarity; density peaks-based clustering





## 1. Introduction

Most complex networks of the real world are composed of network features called community structure, which means that there are vertex groups, in which connections are dense and between which are sparse [1]. By community structure, we can better understand the underlying structure of the community. Community detection methods proposed by scholars usually depend on different standards, such as graph partitioning [2], hierarchical clustering [3], spectral clustering [4], label propagation [5,6], an optimization-based approach [7], and so on. Dao et al. [8] introduced some popular community detection methods that are widely discussed in the literature and made comprehensive analyses on computation time and community size distribution of the mentioned algorithms. The most popular algorithm was introduced by Girvan and Newman [9], and modularity has become the most popular quality function to evaluate community division. The optimization of modularity and the improvement in modularity functions have become two research hotspots.

However, traditional methods based on modularity usually have a resolution limit [10,11]. Social networks with community structure usually contain communities of very diverse sizes, so many small communities may not be detected [12]. Moreover, modularity is extremely sensitive to even individual connections [13]. The emergence of the resolution limit means that the networks may have multi-scale community structures, which make them impossible to rely solely on modularity as a single measurement criterion. Thus, many modularity-based models have an inherent shortcoming, that is, communities smaller than certain scales tend to be merged into adjacent larger ones. In addition, in many instances, to pursue the approximate optimal value of modularity, some algorithms have excessively

divided the community network, causing the result of division to deviate from the real division structure.

Generally speaking, vertices in the same community tend to have more similar attributes and characteristics. As in the recommendation system, the similarity between users is often used to mine and detect user interests [14,15]. From the perspective of node similarity, we can divide the community by the mutual relationships between vertices, which can also be regarded as one of the ideas of community division. Cauteruccio et al. [16] discussed the concept of scope in a Multiple Internet of Things (MIoT). As for the evaluation of user influence, the concept of influence degree in complex networks is similar to the scope of an object in a MIoT. In recent years, the study of community division based on relationships has attracted increasing attention from scholars, and many effective methods have been developed.

In this article, we propose a novel community structure division algorithm, which draws on the idea of a density peaks-based clustering algorithm, called EDPC. Firstly, the central vertex set of the community was determined based on the relationships between vertices, and then each remaining vertex was assigned to the community to which the tightly connected vertices with high relative local density belong. The EDPC algorithm has the following new features: (1) The relative local density is proposed based on the relative connection coefficient between vertices. It narrows the gap of the density of vertices between large-scale and small-scale communities, making smaller-scale communities easier to be discovered. (2) The density peaks-based clustering (DPC) algorithm was applied to preliminary select the central vertices. Then, the central vertices were further filtered according to the connection strength between vertices. (3) No tunable parameters were used in the steps of the algorithm.

The rest of this article is organized as follows. In Section 2, we discuss some related work. In Section 3, we introduce some preliminary concepts of the AA index and the DPC algorithm. Section 4 presents the proposed EDPC algorithm. Section 5 analyses our experiment results. Finally, we summarize this study in Section 6.

## 2. Related Work

Previous related work provided various community definitions and algorithms based on network metrics and structures, and they tried to reveal the communities hidden in these network structures. According to diverse community meta-definitions, Coscia et al. [17] modeled the main aspects of discovering communities and reviewed classical community discovery approaches in recent years. The classification standards of community meta-definitions were feature distance, internal density, bridge detection, diffusion, closeness, structure, link clustering, and meta clustering. In addition, some existing surveys also empirically evaluated the classical community division algorithms. Papadopoulos et al. [18] presented a comparative discussion on the computational complexity and memory requirements of several community detection methods. Harenberg et al. [19] analyzed several community detection algorithms for overlapping and nonoverlapping community detection in large-scale real networks. By comparing their performance of the known ground-truth communities with the structural attributes of the communities identified by these algorithms, it was concluded that some algorithms that could identify communities with "good" structural properties did not necessarily produce good performance indicators.

In a complex network, each vertex does not exist in isolation, and the interrelationship between vertices determines the division of the complex network. Vertex similarity based on network structure information has received extensive attention from scholars. Some of their representative work includes:

Wang et al. [20] presented a BLI algorithm, which employed local neighborhood ratio and degree clustering information to detect the community. The neighborhood ratio was calculated from Jaccard's coefficient. Deng et al. [21] proposed a community detection method by an improved density-peaks model, which estimated composite similarity by normalizing Jaccard and the shortest path feature and acquired key nodes by an improved

density-peaks model with threshold condition. Pan et al. [22] put forward an adjacent node similarity optimization combination connectivity algorithm, which obtained the closet neighboring nodes using local similarity based on the local clustering coefficients. Hoffman et al. [23] introduced a partition algorithm that used Cohen's k as a similarity measure for each pair of nodes in the network, both linked and unlinked. Then, the *k* values were clustered to detect communities. Liu et al. [24] proposed a two-stage BFS local community detection algorithm, which was based on breadth first search, node transfer similarity, and a local clustering coefficient. Sun et al. [25] designed a community detection method based on the Matthew effect. The authors used the degree of nodes and the Jaccard's coefficient between them to describe the attraction of nodes, which were used to form the core groups. Each core group constantly attracted peripheral nodes to join and form a larger community structure until the whole network reached a stable state. Jiang et al. [26] introduced a community detection algorithm for complex networks (CLPE) based on a link prediction strategy to detect community structure, especially for networks with unclear structure. Agrawal et al. [27] proposed an unsupervised graph clustering algorithm (SAG-Cluster) with K-medoids, which used collaborative similarity measures based on a distance function to detect communities.

## 3. Preliminaries

Suppose that the undirected and unweighted social network is modeled as a graph $G = (V, E)$, where each vertex in $V$ represents a social network user, and each edge in $E$ represents a link between two network users. The adjacent matrix of the graph $G$ is $A$, and $A_{ij}$ is the value in the *i*th row and the *j*th column of matrix $A$. If there is an edge between vertices *i* and *j*, $A_{ij}$ is equal to 1, and the pair of vertices are neighbors. Otherwise $A_{ij}$ is equal to 0. Obviously, the adjacency matrix of an undirected graph is always symmetric. The number of vertices is denoted by $|V| = N$, and the number of edges is $|E| = m$; $k_i$ is defined as the degree of the vertex $v_i$.

### 3.1. Adamic–Adar Index

In social networks, there are usually certain similarities between pairs of vertices, which can be used to measure whether two vertices belong to the same community or not. Following different indexes, the similarity of different aspects of vertices can be measured. From the perspective of network topology, the degree of common neighbors (CN) [28] measures the number of shared neighbors of two vertices, which is the easiest way to define similarity. Inspired by the CN index, increasing the influence of the vertex degree, other normative similarity indexes can be obtained, such as the Salton index (SAL) [29], Jaccard's coefficient (JC) , the Adamic–Adar (AA) index [30], the Resource Allocation index (RA) [31], etc.

Among the above-mentioned indexes, the Adamic–Adar index refines the simple counting of common neighbors by assigning the lower-connected neighbors' higher weight. The AA index is formulated as follows.

$$S_{ij} = \sum_{z \in \Gamma(i) \cap \Gamma(j)} \frac{1}{\log k(z)} \tag{1}$$

where $\Gamma(i)$ is defined as the neighbor set of vertex *i*. $k(i)$ is the degree of vertex *i*, which is also defined as $k(i) = |\Gamma(i)|$.

In fact, the importance of each neighbor is different. Sometimes, the fewer the neighbors a vertex has, the more important it is as an intermediate vertex. The AA index takes into consideration not only the degree of common neighbors but also the contribution of common neighbors with different degrees. For vertices *i* and *j*, the influence of each common neighbor on the relationship between the pair of vertices is not exactly the same. Through the analysis, it can be found that using the AA index, in the set of common neighbors, the contribution of vertex with a smaller degree is greater than that of the vertex with a larger degree.

### 3.2. Density Peaks-Based Clustering

Density peaks-based clustering (DPC) [32] is based on the assumptions that a cluster center has two characteristics: one is that the center point's local density is higher than that of neighbors, and the other is that the center point has a relatively large distance from points with higher local density. Given a set of data points, DPC defines the local density $\rho_i$ of data point $i$ according to

$$\rho_i = \sum_j \chi(dist(i,j) - d_c) \tag{2}$$

$$\chi(x) = \begin{cases} 1, & if\ x < 0 \\ 0, & otherwise \end{cases} \tag{3}$$

where $dist(i,j)$ is the distance between point $i$ and $j$, and $d_c$ is a cutoff distance.

$\delta_i$ is measured by computing the minimum distance between the point $i$ and any other point with higher density, which is defined as

$$\delta_i = \begin{cases} \max_j(d_{ij}), & otherwise \\ \min_{j:\rho_j > \rho_i}(d_{ij}), & if\ \exists j\ s.t.\ \rho_j > \rho_i \end{cases} \tag{4}$$

For the point with highest density, we normally take $\delta_i = max_j(dist(i,j))$. Those centers of clusters simultaneously have relatively high $\delta$ and $\rho$. If we define $\gamma_i = \rho_i \delta_i$, it also means that the value of $\gamma$ for cluster centers are relatively high. In order to avoid the influence of statistical errors of cases with few points, one of the improvements is to use a Gaussian kernel to adjust the local density, which is defined by

$$\rho_i = \sum_j \exp(-\frac{dist(i,j)^2}{d_c^2}) \tag{5}$$

## 4. The Proposed Algorithm

To extend DPC algorithm to community partition and to avoid the resolution limitation, we proposed an extended density peaks-based clustering, named EDPC. The main features of the EDPC algorithm are as follows: (1) it determines the connection strength between vertices on the basic of AA index; (2) it calculates the local density of each point based on the connection coefficient; (3) according to the DPC algorithm, candidate community centers are selected; (4) it selects final community centers according to the connection strength, then the remaining vertices are assigned into corresponding sub-communities. In this section, we discuss our work, which are the key parts of the EDPC algorithm.

### 4.1. Connection Strength

In real social relations, two people who have more mutual friends tend to have higher similarities, and they are more likely to belong to the same community. Using the Salton index, Jaccard's coefficient, the Adamic–Adar index, we can estimate the node similarity of a social network. However, for some networks with special structures, the description of the basic node similarity index is not very accurate. We take the network in Figure 1 as an example. If the AA index is used to calculate the similarities between vertices, the value of similarity between vertex 2 and vertex 3 is $2/\log 3$, and vertex 1 and vertex 5 also have a similarity of $2/\log 3$. However, since there is an edge between vertex 2 and vertex 3, theoretically, the compactness of the former should be higher than that of the latter. Apart from this, in Figure 1, using the AA index, the similarity between vertex 1 and 2 is the same as the similarity between vertex 2 and 5. Analyzing the network structure, we found that the degree of vertex 1 was obviously smaller than that of vertex 5. From the perspective of maintaining the relationship, to maintain the relationship with vertex 2, the investment of vertex 1 was greater than that of vertex 5. This means that the relationship between vertices 1 and 2 was more closely related.

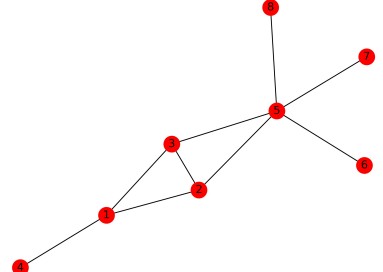

**Figure 1.** Example of an undirected network.

Therefore, based on the AA index, the adjacency matrix, and the vertex degree, we reconstructed the formula of connection strength between vertex pairs. The connection strength between two vertices can be defined as

$$CS_{i,j} = S_{ij} + \frac{A_{ij}}{\max(k(i), k(j))} \tag{6}$$

The use of $\max(k(i), k(j))$ in the formula is a consideration based on people's behavior; that is, the more friends a person has, the less he invests in maintaining each friend's relationship on average. Correspondingly, his average intimacy with each friend is also low. It should be noted that $CS_{i,j}$ is a local similarity indices describing the tightness of vertices. When vertices $i$ and $j$ are neither directly connected nor have a common neighbour, the value of $CS_{i,j}$ is still 0. By Equation (6), we can calculate that the value of $CS_{2,3}$ is greater than $CS_{1,5}$, and the value of $CS_{1,2}$ is greater than $CS_{2,5}$, which are reasonable from the topological structure of the network.

*4.2. Relative Connection Coefficient*

In a social network, a community center will have stronger cohesion than its neighboring vertices. The sum of the connection strength of a vertex and its neighbors can be considered as the connection coefficient of this vertex. However, many community detection algorithms based on modularity methods usually have a resolution limitation that makes them unable to detect communities that are sufficiently smaller compared to the entire network [10,33]. To overcome the resolution limitation caused by the large gap in community size, we proposed to define the relative connection coefficient of each vertex based on the ratio of a vertex's connection coefficient to the sum of its neighbors' connection coefficients. The connection coefficient of vertex $i$ is defined as

$$CC_i = \sum_{j \in \Gamma(i)} CS_{i,j} \tag{7}$$

And the relative connection coefficient of vertex $i$ is defined as

$$RCC_i = \exp\left(\frac{CC_i * k(i)}{\sum_{j \in \Gamma(i)} CC_j}\right) \tag{8}$$

A social network with a community structure usually contains communities of various sizes. In a large-scale community, the connection coefficient of each vertex is often larger. On the contrary, the connection coefficient of each vertex in a small-scale community is relatively weak. This phenomenon makes it difficult to detect the central vertex of a small-scale community. To a certain extent, the relative connection coefficient, which considers the vertex degree and the connection coefficient of each neighbor vertex, balances this gap. Through this strategy, even if there are large differences in the size of the communities in the network, the central vertices in the small-scale communities will become more prominent due to their higher relative connection coefficient.

### 4.3. Community Centers

In the actual complex network, there is such a situation that the vertices in the same community are closely related, and the central vertices of different communities are relatively far away. Therefore, in the EDPC algorithm, relative connection coefficient of vertex $i$ was utilized as local density $\rho_i$, which is conductive to the discovery of small-scale communities.

The calculation of distance $\delta_i$ is relatively simple: the shortest path length between vertex $i$ and any other vertex with higher density. We computed $\delta_i$ using Equation (4), where the distance $d_{ij}$ between vertex $i$ and vertex $j$ in a network is the number of edges in the shortest path connecting them, namely, geodesic distance. For the vertex $i$ with the highest local density, we took $\delta_i = \max_j(d_{ij})$, which means the eccentricity of a vertex with the highest $\rho_i$ is the greatest distance between the vertex $i$ and any other vertex.

According to the DPC algorithm, the community centers are considered as vertices with relatively high $\rho_i$ and high $\delta_i$. Correspondingly, there will be a higher $\gamma_i$, where $\gamma_i = \rho_i \delta_i$. We sorted the local density of each vertex in descending order and calculated the distance of each vertex; then, a decision diagram was constructed. Furthermore, another decision diagram was drawn according to the value of $\gamma$. After referring to the two diagrams, the vertices with higher $\rho$, $\delta$, and $\gamma$ were selected as the candidate community centers.

It was found that if a social network has a small standard deviation of the local density, the vertex positions in two decision diagrams will be denser and difficult to distinguish. For a network with such a topological structure, it is difficult to distinguish the candidate center set manually. In view of this problem, we arranged the local densities in descending order. Then, we selected the vertices with local density values greater than $\lambda$. We calculated $\lambda$ using Equation (9).

$$\lambda = Std(RCC) + \frac{Mean(RCC)}{2} \tag{9}$$

$Std(RCC)$ and $Mean(RCC)$ represent the standard deviation and average value of $RCC$ of all vertices, respectively. It should be noted that Equation (9) is necessary only when the decision diagrams cannot clearly distinguish candidate vertices.

After each candidate center is arranged in descending order of $\gamma$, the candidate center with the highest $\gamma$ is defined as the first determined center. We sequentially calculated the connection strength between each candidate center and the determined center with a higher $\gamma$ value, and we determined whether it is the next determined center or in the same community with the existing determined center. If the connection strength from a candidate center to one of the determined centers is relatively high, they are likely to belong to the same community, so this candidate center will be removed from the candidate center set. Otherwise, this candidate center is defined as the next determined center of another community. In our proposed method, the criterion for excluding a candidate center is whether the value of $CS_{i,k}$ is greater than half of $maxCS_i$, where vertex $k$ is the candidate center waiting for further judgment, and vertex $i$ is any determined center vertex with the higher value of $\gamma$. Through the analysis of networks with different structures, we found that the central vertices satisfying this criterion were usually located in different subcommunities. We computed $maxCS_i$ using Equation (10) as follows and repeated this judgment on each candidate center.

$$maxCS_i = \max(CS_{i,j}) \quad where \; j = 1, \ldots, N \; and \; i \neq k \tag{10}$$

Here, we used a simple network with 16 vertices and 36 edges to demonstrate the selection of central vertices in detail. The example network is shown in Figure 2. According to Equations (4) and (8), we calculated $\rho$, $\delta$ and $\gamma$ of each vertex and drew two decision diagrams. Vertex 13 had the maximum local density value of 3.814, and vertices 15 ($\rho_{15} = 3.698$, $\gamma_{15} = 7.397$) and 7 ($\rho_7 = 3.130$, $\gamma_7 = 6.260$) also had higher values of $\rho$ and $\gamma$. Figure 3a shows the plot of $\delta_i$ as the function of $\rho_i$ for each vertex, and the positions of

vertices 13, 15, and 7 were obviously far away from other points. In Figure 3b, we also find the $\gamma$ maxima were at these vertices, which we identified as candidate centers.

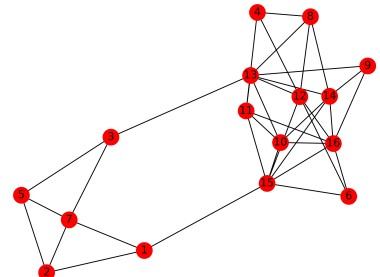

**Figure 2.** Example of another undirected network.

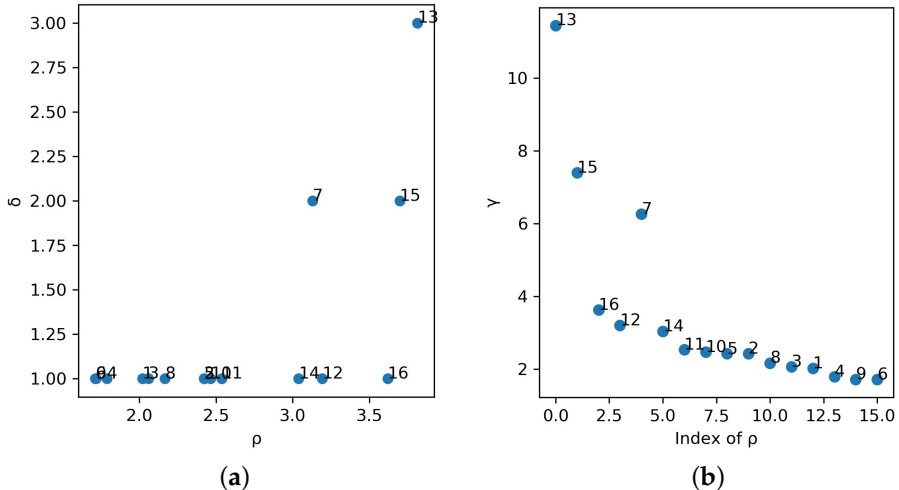

**Figure 3.** Decision diagrams for the example network. (**a**) Decision diagram for the vertices. (**b**) $\gamma$ distribution.

It can be calculated that the average value of $\rho$ was 2.613, and the standard deviation of $\rho$ was 0.693. Although vertices 16 ($\rho_{16} = 3.621$, $\gamma_{16} = 3.621$), 12 ($\rho_{12} = 3.194$, $\gamma_{12} = 3.194$), and 14 ($\rho_{14} = 3.040$, $\gamma_{14} = 3.040$) had larger values, they were excluded from the candidate center set because their corresponding distances $\delta$ were both 1. We arranged these vertices in the candidate center set in descending order of $\gamma$, where the order was 13, 15, 7.

Because vertex 13 had the $\delta$ maxima and the $\rho$ maxima, we determined it as one of the centers and calculated the connection strength between vertex 13 and 15. Since $CS_{13,15}$ was greater than $maxCS_{13}/2$ ($CS_{13,15} = 2.359$, $maxCS_{13} = 3.269$), it means that they were closely related to each other and should be in the same community. Vertex 15 was deleted from the candidate center set. Because the connection strength between vertices 13 and 7 was relatively small ($CS_{13,7} = 0.910$), it means that they are less compact and should be in different communities, so vertex 7 should also be one of the determined centers. Finally, the determined central vertices of the example network were vertices 13 and 7.

*4.4. Division of Remaining Vertices*

After the community centers were finally selected, each remaining vertex was classified into the same subcommunity where the nearest and denser vertices are located. However, through the analysis of the actual community division, we noticed that such a situation inevitably exists in the actual division process; that is, the distances to multiple vertices with high local density from a remaining vertex may be exactly identical. Therefore, we need to further consider how to classify this special remaining vertex.

In descending order of $\gamma$, we first calculated the shortest distance and connection strength from each remaining vertex to all vertices whose local density was higher than it. If there was only one higher local density vertex with a shortest distance of 1, then the remaining vertex was consistent with its category. Otherwise, if there were multiple vertices with higher local density that are one step away from this remaining vertex, and these vertices have been divided into different subcommunities, the EDPC algorithm will calculate $sumCS_{r,k}$ with Equation (11). The remaining vertex was assigned to the subcommunity with the maximum in $sumCS$.

$$sumCS_{r,k} = \sum_{j:\rho_j > \rho_i \ and \ j \in Com_k} CS_{r,j} \qquad (11)$$

where $r$ represents the index of a remaining vertex, $k = 1, \ldots, Num$; where $Num$ represents the number of the subcommunities. Here, $j$ is the vertex with higher local density and belongs to the $k$th subcommunity.

If the shortest path from the higher local density vertex to a remaining vertex is a two-step distance, the allocation approach is consistent with that of the one-step distance. Since connection strength can only describe the relationships of vertices within the two-step distance, when the distance is greater than 2, the remaining vertex is classified into the subcommunity where the vertex with maximum local density is located.

### 4.5. EDPC Method

The pseudo-code of the EDPC algorithm is presented in Algorithm 1.

---

**Algorithm 1** EDPC Algorithm.

---

**Require:** Adjacency matrix $A$ of a network $G$
**Ensure:** The allocation for all date vertices
1: Calculate $CS_{i,j}$ via Equation (6);
2: $d_{ij} \leftarrow$ shortest path distance between vertices $i$ and $j$;
3: Calculate $\rho_i$ and $\delta_i$ via Equations (4) and (8);
4: $\gamma_i \leftarrow \rho_i \delta_i$;
5: Preliminary selection of candidate community centers;
6: Rescreen the community centers;
7: Allocate the remaining vertices;

---

Our algorithm consists of two main steps. The first step is to determine the central vertices of the community, and the second step is to allocate the non-center vertices. Suppose there is a network with $N$ vertices and the number of the subcommunities is $c$. The time complexity of EDPC is as follows.

For the first step, its complexity includes the following sub-steps: computation of connection strength with the time complexity of $O(N^2)$ and computation of $\rho$ with the time complexity of $O(N^2)$. As for selecting central vertices, sorting $\gamma$ requires $O(NlogN)$, and selecting centers requires $O(N)$ in the worst case. For the second step, the time complexity of allocating the remaining vertices is $O(cN)$. Thus, the total time of complexity of our method is $O(2N^2 + NlogN + N + cN)$.

## 5. Experiments and Result Analysis

In this section, we first briefly introduce the popular community partition algorithms, datasets, and evaluation metrics. Then, we analyze the experimental results obtained by EDPC and other baseline algorithms on real-world networks and synthetic networks.

The compared methods are as follows: Fastgreedy [34], BGLL [35], Walktrap [36], Meme-net [37], Optimal modularity [38], Spinglass [39], and LPA [5]. The corresponding functions provided by the Python-igraph library were used to compare the division results and the performance of the algorithms. In meme-net algorithm, because the value of parameter $\lambda$ will affect the number of divided communities, a compromise is to define

$\lambda$ as 0.5. We utilized two evaluation metrics to compare the effect of community division, namely, the normalized mutual information (NMI) [12] and the adjusted rand index (ARI) [40]. NMI and ARI are two widely used evaluation metrics, which are applied to measure the similarity between the community division obtained by the algorithm and the real community division, so as to objectively estimate the accuracy of a community division compared with the standard division. These two indexes are bounded above by 1. The closer the value is to 1, the more similar the division result is to the real community division. For BGLL, Meme-net, Spinglass, and LPA, we repeated 30 independent runs to calculate the average values of NMI and ARI.

*5.1. Datasets*

To assess the performance of EDPC, we chose four real-world social networks and two synthetic networks based on LFR benchmarks [41]. The four familiar real world-networks were Zachary's Karate Club, the Dolphins Social Network, American Football Clubs, and Books about US Politics, which are usually used to test the effectiveness of community partition algorithms. Another major reason for choosing these networks is that the ground truth about them is known, so it is convenient for us to evaluate the results of the comparison algorithms. These real-world networks are compiled by Newman (Available at http://www-personal.umich.edu/~mejn/netdata/, accessed on 25 November 2021). The topological properties of the four real-world networks are shown in Table 1. Here, $d_{avg}$ denotes the average of the vertex degree, $k_{avg}$ denotes the average of the local clustering coefficient, and $c$ denotes the number of subcommunities.

**Table 1.** Attributes of four real-world networks.

| Networks | Vertices | Edges | Density | $d_{avg}$ | $k_{avg}$ | Avg. Path Length | c |
|---|---|---|---|---|---|---|---|
| Karate | 34 | 78 | 0.139 | 4.588 | 0.571 | 2.408 | 2 |
| Dolphins | 62 | 159 | 0.084 | 5.129 | 0.259 | 3.357 | 2 |
| Football | 115 | 613 | 0.094 | 10.661 | 0.403 | 2.508 | 12 |
| Polbooks | 105 | 441 | 0.081 | 8.400 | 0.488 | 3.079 | 3 |

In addition, two synthetic LFR networks were used in experiments, whose numbers of vertices were 500 and 1000, respectively. The basic topological properties of each LFR network are shown in Table 2. The parameter $\mu$ means mixing parameter. The higher the $\mu$ of a network is, the more difficult it is to reveal the community structure. On the contrary, the smaller the $\mu$, the more compact the community is. In LFR1 network and LFR2 network, we defined the parameters as 0.5 and 0.4, respectively. Therefore, although the scale of these two networks was not very large, it is difficult to divide them. The LFR1 network contains 18 subcommunities, and the LFR2 network contains 16 subcommunities.

**Table 2.** Parameter setting of two LFR networks.

| Networks | Vertices | $k$ | $k_{max}$ | $\mu$ | $C_{min}$ | $C_{max}$ |
|---|---|---|---|---|---|---|
| LFR1 | 500 | 10 | 20 | 0.5 | 10 | 50 |
| LFR2 | 1000 | 10 | 20 | 0.4 | 30 | 90 |

*5.2. Experiments on Datasets and Results Analysis*

In this subsection, we focus on the performance of EDPC, Fastgreedy, BGLL, Walktrap, Meme-net, Optimal modularity, Spinglass, and LPA on datasets listed in Tables 1 and 2. As can be seen from Table 3, the partition result of EDPC was basically consistent with reality. For the Karate network, the Dolphins network, and the LFR2 network, the number of subcommunities divided by our method was the same as the number of real communities. The number of divided communities in the remaining networks was also close to the real number.

By observing Table 3, we noticed that the maximum values of NMI obtained by the EDPC method on four real networks were 1, 1, 0.9197, and 0.6096, respectively. Compared with other comparison algorithms, using the EDPC algorithm, all four real-world networks obtained the highest NMI values. Furthermore, as shown in Table 4, in terms of the benchmark ARI, our algorithm outperformed all other seven algorithms. For the Football network and the Polbooks network, although the number of subcommunities divided by the EDPC algorithm was inconsistent with the actual number of communities, the values of NMI and ARI corresponding to the two networks were all the highest, which shows that the EDPC algorithm is outstanding in terms of community division.

Analyzing the experimental results in Tables 3 and 4, we can find that for the network with more vertices (such as LFR1 and LFR2), the Optimal modularity algorithm cannot obtain the partition result in the effective time. The mixing parameter of the LFR1 network was 0.5, and the larger mixing parameter interfered with the LPA algorithm. However, the EDPC algorithm still performs well when the structure of synthetic network is not clear enough. In the LFR1 network, the EDPC algorithm acquired the best NMI and ARI values. In the LFR2 network, the value of NMI obtained by the EDPC algorithm was slightly lower than the Spinglass algorithm, but the value of ARI was higher than other comparison algorithms. The experimental results show that our algorithm was more conductive to discovering the central vertices of the communities and identifying the actual community structure.

**Table 3.** Results of NMI and the number of communities on networks.

|  | Karate | Dolphins | Football | Polbooks | LFR1 | LFR2 |
|---|---|---|---|---|---|---|
| BGLL | 0.7105 (4) | 0.5716 (5) | 0.8752 (9/10) | 0.5380 (5) | 0.6772 (12) | 0.8846 (15) |
| Fastgreedy | 0.8168 (3) | 0.6020 (4) | 0.7081 (6) | 0.5313 (4) | 0.3928 (6) | 0.4789 (6) |
| Walktrap | 0.5309 (5) | 0.5652 (4) | 0.8879 (10) | 0.5437 (4) | 0.8104 (16) | 0.9141 (16) |
| Meme-net | 0.6790 (3) | 0.5785 (5) | 0.8962 (11) | 0.4812 (5) | 0.6286 (11) | 0.8279 (15) |
| OM [38] | 0.6187 (4) | 0.6441 (5) | 0.8910 (10) | - | - | - |
| Spinglass | 0.6557 (4) | 0.6400 (5/6) | 0.9019 (10–12) | 0.5091 (6) | 0.9453 (14) | **0.9903 (17/18)** |
| LPA | 1 (2) | 0.6686 (4/5) | 0.8733 (9–12) | 0.5559 (3/4) | - | 0.8369 (12) |
| EDPC | **1 (2)** | **1 (2)** | **0.9197 (10)** | **0.6069 (2)** | **0.9579 (15)** | 0.9896 (16) |

**Table 4.** Results of ARI on real-world networks.

|  | Karate | Dolphins | Football | Polbooks | LFR1 | LFR2 |
|---|---|---|---|---|---|---|
| BGLL | 0.5547 | 0.3475 | 0.7784 | 0.6131 | 0.4432 | 0.7910 |
| Fastgreedy | 0.7411 | 0.4509 | 0.4741 | 0.6379 | 0.1576 | 0.2517 |
| Walktrap | 0.3207 | 0.4167 | 0.8154 | 0.6534 | 0.6429 | 0.8822 |
| Meme-net | 0.4921 | 0.3511 | 0.8043 | 0.4906 | 0.4387 | 0.7782 |
| OM [38] | 0.4646 | 0.3735 | 0.8069 | - | - | - |
| Spinglass | 0.4884 | 0.3682 | 0.8134 | 0.5068 | 0.8642 | 0.9883 |
| LPA | 1 | 0.4351 | 0.7362 | 0.648 | - | 0.4556 |
| EDPC | **1** | **1** | **0.8406** | **0.6671** | **0.8969** | **0.9899** |

Since the EDPC algorithm first determines the central vertices and then divides the remaining vertices, we can dig out the key vertices in a social network while dividing the network. For clarity, we chose the Football network as an instance. Due to the particular topology of the Football network, many classical algorithms often cannot accurately determine its central vertices. Therefore, we elaborated the connection strength between the central vertices of the Football network and visualized the detected communities. The vertices were drawn in different colors corresponding to their communities.

With reference to the center selection criterion of EDPC algorithm, we finally chose 10 vertices as community centers, and the connection strength between these vertices is shown

in Table 5. In Table 5, we list the maximum connection strength of each central vertex and the connection strength between each pair of central vertices. Through specific data, we can easily find that the connection strength between each pair of central vertices was far less than the maximum connection strength of the two central vertices. The results for real network partition and our algorithm's partition are given in Figure 4a,b, respectively.

**Table 5.** Connection strength of community centers on Football network.

| Vertex | $maxCS$ | 6 | 2 | 19 | 7 | 89 | 77 | 1 | 8 | 67 | 20 |
|---|---|---|---|---|---|---|---|---|---|---|---|
| 6 | 3.040 | - | 0 | 0 | 0 | 0.434 | 0 | 0.872 | 0 | 0 | 0 |
| 2 | 3.440 | 0 | - | 0 | 0 | 0.417 | 0.417 | 0.518 | 0 | 0 | 0.869 |
| 19 | 3.176 | 0 | 0 | - | 0 | 0 | 0 | 0 | 0 | 0 | 0.572 |
| 7 | 2.905 | 0 | 0 | 0 | - | 0.434 | 0 | 0 | 0.083 | 0 | 0.417 |
| 89 | 3.408 | 0.434 | 0.417 | 0 | 0.434 | - | 0 | 0 | 0.417 | 0 | 0 |
| 77 | 2.971 | 0 | 0.417 | 0 | 0 | 0 | - | 0 | 0 | 1.006 | 0 |
| 1 | 3.061 | 0.872 | 0.518 | 0 | 0 | 0 | 0 | - | 0 | 0.417 | 0.851 |
| 8 | 3.771 | 0 | 0 | 0 | 0.083 | 0.417 | 0 | 0 | - | 0 | 0 |
| 67 | 2.954 | 0 | 0 | 0 | 0 | 0 | 1.006 | 0.417 | 0 | - | 0.417 |
| 20 | 3.045 | 0 | 0.869 | 0.572 | 0.417 | 0 | 0 | 0.851 | 0 | 0.417 | - |

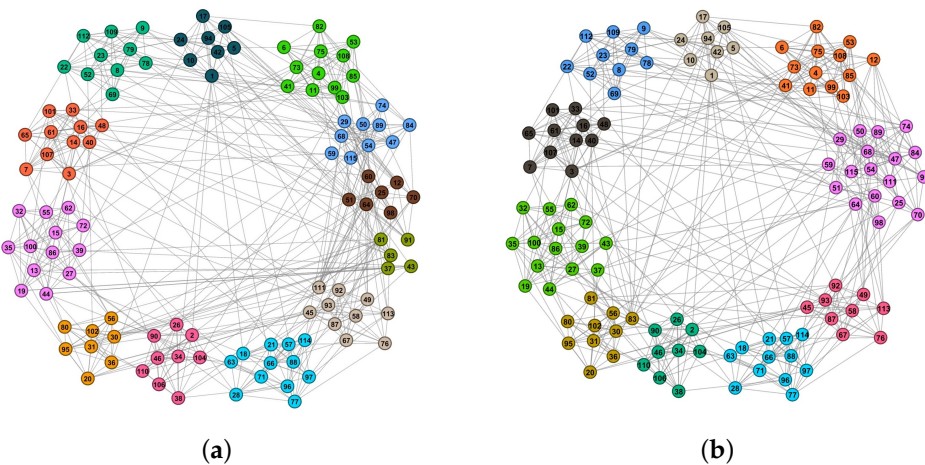

    (a)         (b)

**Figure 4.** The community structure of Football network. (**a**) The real community structure. (**b**) The community structure detected by EDPC.

The comparison between the results of EDPC and the other three algorithms in the Dolphins network, the Football network, and the Polbooks network is depicted in Figure 5. The results of 10 independent repeated experiments of different algorithms on the same network reflect the performance of algorithms in community detection. As Figure 5 shows, when the other three comparison algorithms divide the network, the structure and performance of the division will produce unpredictable fluctuations. However, the EDPC algorithm has the characteristics of first evaluating the community vertices and then dividing them, so that the EDPC algorithm can ensure a constant division result. It can also be observed from Figure 5, in the three real-world networks, that the EDPC algorithm achieved the optimal value of NMI. Although the BGLL algorithm obtained the optimal result of ARI in the Football network and the LPA algorithm obtained the optimal result of ARI in the Polbooks network, due to the fluctuation of the results of these two algorithms, the average values of ARI were still lower than the average value of the EDPC algorithm.

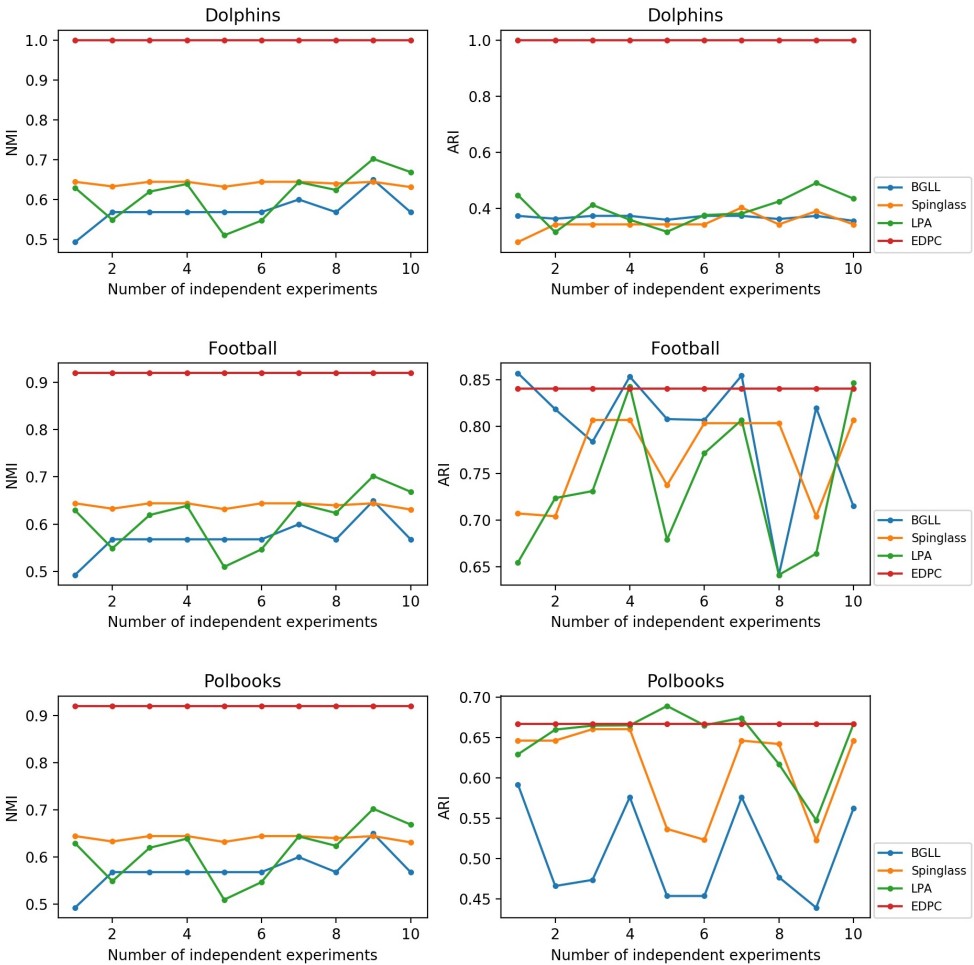

**Figure 5.** Comparison of NMI and ARI.

## 6. Conclusions

In this study, we explored the problems of finding the central vertices of each community and discovering community structure in social networks. In the EDPC algorithm, the AA index was presented to measure the connection strength between vertices, and then the local density and distance of each vertex were calculated. According to the basic idea of the DPC algorithm and other constraints, the initial centers were selected first. This approach can ensure that the true centers are not missed. In the subsequent steps, by comparing the strength connection between each center, the false centers were picked out. Finally, referring to the divided vertices, the remaining vertices were sequentially assigned to the most closely connected communities. Experiments on several real-world networks and synthetic networks proved that the EDPC is stable and that the division results are reasonable. Our proposed method starts from the network structure and pursues the real network by exploring the internal relationship between vertices. Therefore, our algorithm provides an alternative method to discover real network partition.

However, there are still some potential problems that need to be investigated further. In future work, we will focus on extending the proposed algorithm to the weighted and directed social networks. In addition, we will compare the impact on the performance of the EDPC algorithm based on different node similarity measurements.

**Author Contributions:** Two authors contributed to this work. Methodology, Y.M. and X.L.; original draft preparation Y.M.; writing-review and editing, Y.M. and X.L.; supervision, X.L. All authors have read and agreed to the published version of the manuscript.

**Funding:** This research was supported by the National Nature Science Foundation of China (grant numbers 61876101, 61802234, and 61806114), the Humanities and Social Sciences Project of Education Ministry (grant number 20YJA870013), and the Natural Science Foundation of Shandong Province (grant number ZR2019MF016).

**Institutional Review Board Statement:** Not applicable.

**Informed Consent Statement:** Not applicable.

**Data Availability Statement:** Not applicable.

**Conflicts of Interest:** The authors declare no conflict of interest.

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
