# Peer review of "Finding Central Vertices and Community Structure via Extended Density Peaks-Based Clustering"

_information, doi:10.3390/info12120501_

Round 1
Reviewer 1 Report
This manuscript propose a method to filter community centers by the relative connection coefficient between vertices and analyze the
community structure accordingly. The paper is technically sound, and well written, the proposed method is well explain, and the conclusions are supported by the presented experiment and results. However, i do have few minor comments and suggestions:
- Add a related work section and move the following content from the Introduction: from line 41 " Wang et al. [13]" to the end of line 62 should be moved to the related works section.
-The introduction should be extended (it would be too short after moving most of content to the related work section, as required in the last comment)
- There is mistake in the title of section 2, it should be preliminaries, not Related works.
- The writing in Figure 5 is too small and not clear at all, which makes it difficult to read the values of the compared graphs.
-What is 'Number' in the x-axis labels in Figure 5, number of what?
-Add the following works to the related work section:
[1] A comparative evaluation of community detection methods." Network Science 8, no. 1 (2020): 1-41.
[2] ComPath: User interest mining in heterogeneous signed social networks for Internet of people. IEEE Internet of Things Journal, 2020 , 8(8), pp.7024-7035.
[3] Personality-Aware Product Recommendation System Based on User Interests Mining and Metapath Discovery." IEEE Transactions on Computational Social Systems 8, no. 1 (2020): 86-98.
Reviewer 2 Report
In the proposed manuscript, the authors propose a community structure division algorithm based on a density peaks-based clustering algorithm. The algorithm works by first selecting initial centers of a community. Then, these are filtered out w.r.t. the strength of their connections. The remaining vertices are then assigned to the most closely connected communities. The authors carry out an experimental evaluation involving a comparison with other well known approaches.
The manuscript is well written and it flows well. It is technically sound, although few details are missing. The experiments are solid and show a consistent efficacy of the proposed algorithm. However, in order to be considered for publication, the quality of the manuscript could be improved. I suggest the authors to take into account the following points to improve the manuscript:
- Eq. 6 should be only defined for pairs of nodes <i,j> having a connection; if both nodes are disconnected, the formula is undefined.
- In Eq. 10 the rationale behind the choice of the constants should be provided clearer.
- Page 6: "It is generally believed that the compactness" to support such claim, the compactness should be quantified, although it is possibile to understand what the authors mean in this context. However, I suggest to make this claim clearer or to define the compactness and show why the claim is commonly used. Also, a reference could be used.
- A complexity analysis of the EDPC algorithm is missing. The authors should include it.
- What is the rationale behind the choice of NMI as evaluation metric?
- The relative connection coefficient is similar to the concept of scope in complex networks; the authors could consider citing the following pointer: "An approach to compute the scope of a social object in a Multi-IoT scenario." Pervasive and Mobile Computing 67 (2020): 101223.
Minor issues are highlighted in the following:
- Eq. 1: S_xy should be S_ij
- Eq. 4 is not clear; if the authors intend to say "j such that \ro_j > \ro_i" I suggest to use the set notation instead; the same applies in Eq. 9
- Eq. 8: in the denominator, CC_j should be the body of the summation
- Line 234: please provide the link to the website.
- The authors could report the density value for each network used in the experiments.
- Few typos and repetition of thoughts are present in the manuscript. I suggest the authors to fix them.
Reviewer 3 Report
The paper proposes a new method to filter community centers by the relative connection coefficient between vertices in a network.
As a general comment, I think the paper makes an interesting contribution to the literature by providing this new algorithm. At the same time, I think the paper needs some minor improvements before being published in this journal.
Major comments:
- I suggest the authors call section 2 something like Preliminary concepts and create a new Section “Related work” where the literature review on the social network analysis is addressed.
- In section5 conclusion, I would suggest the authors add a discussion about the practical implication of the proposed algorithm.
Round 2
Reviewer 2 Report
The authors successfully addressed my concerns. The quality of the paper has been massively improved, the fundaments are now more rigorous and well explained. I suggest the manuscript for publication.
Author Response
With your rigorous opinions, the quality of the manuscript has been improved. Under your guidance, I also have an in-depth understanding of manuscript writing. Thank you again for your selfless help.